# *Competitive Coherence* Generates *Qualia* in Bacteria and Other Living Systems

**DOI:** 10.3390/biology10101034

**Published:** 2021-10-12

**Authors:** Vic Norris

**Affiliations:** Microbiology Signals and Microenvironment, Université de Rouen, 76821 Mont Saint Aignan, France; victor.norris@univ-rouen.fr

**Keywords:** consciousness, qualia, network, bacteria, learning, phenotype

## Abstract

**Simple Summary:**

Subjective experiences, sensations, or feelings, *alias qualia*, include the experiences of seeing the colour blue, hearing the sound of middle C, smelling a rose, touching an ice cube and feeling a pain in one’s knee. The origin of these *qualia* and their relationship with the material world have long been subject to intense debate. Here, I propose that *qualia* arise from—and can even determine—the operation of *competitive coherence*. The idea behind competitive coherence is that (1) the behaviour of a system depends on the activity of only a subset of the elements of the system and (2) the selection of elements belonging to this subset depends on both the history and the present state of the system (which includes its present environment). In arguing that competitive coherence is characteristic of living systems at every level, I contend that it is to bacteria that we should turn if we are to understand *qualia*. This is because bacteria were here first, constitute much of the biomass, are very complex, made the world as we know it, and help control its denizens. Finally, I propose that *qualia* are important in the functioning of high-level systems such as ecosystems.

**Abstract:**

The relevance of bacteria to subjective experiences or *qualia* is underappreciated. Here, I make four proposals. Firstly, living systems traverse sequences of *active* states that determine their behaviour; these states result from *competitive coherence*, which depends on connectivity-based competition between a *Next* process and a *Now* process, whereby elements in the active state at time *n+1* are chosen between the elements in the active state at time *n* and those elements in the developing *n+1* state. Secondly, bacteria should help us link the mental to the physical world given that bacteria were here first, are highly complex, influence animal behaviour and dominate the Earth. Thirdly, the operation of competitive coherence to generate active states in bacteria, brains and other living systems is inseparable from *qualia*. Fourthly, these *qualia* become particularly important to the generation of active states in the highest levels of living systems, namely, the ecosystem and planetary levels.

## 1. Introduction

Subjective experience has an intimate relationship with “consciousness” [1]. Indeed, it is generally considered that the problem surrounding consciousness is to explain how and why subjective, *felt* states or *qualia* exist [2], with *qualia* taken to be what is primary and most fundamental about our consciousness [3]. These *qualia* are considered the intrinsic properties of subjective experiences and constitute the mysterious essence of particular sensations such as blueness, heat, etc. [4]. In his influential review [2], Chalmers draws a distinction between the easy and hard problems in consciousness studies. The easy problems all concern the performance of various functions, which are explained once the mechanics of this performance has been explained. In contrast, the hard problem is not a problem related to how functions are performed; once the performance has been explained, the question remains as to why the performance of this function is associated with subjective experience. 

In grappling with this hard problem, it is often argued about how far subjective experience extends “down the phylogenetic scale” [5,6,7,8,9,10]. In vertebrates, the capacity for subjective experience is proposed to be due to integrated structures in the midbrain that create a neural simulation of the state of the mobile animal in space whilst similar structures in the insect brain perform similar functions and may provide a similar capacity [11] (however, see [12]). In plants, arguments have been made both in favour of consciousness [6] and against [8]. In thinking about a possible evolutionary relationship between phylogeny and subjective experience, it is sometimes assumed firstly that evolution is accompanied by increasing complexity (such that life ascends through “grades of organisms, with divergently increasing complexity of organic structure and correlated … mental or psychical complexity” [13]) and secondly that exceeding a threshold of complexity is needed for subjective experience; that is, subjective experience evolved from its absence in “simple” organisms to its presence in other organisms of “greater complexity” [1]. *Unlimited associative learning* is thought to allow an organism that has passed this threshold to be motivated by a novel stimulus or action that can then serve as the basis for learning and that constitutes the basis of *qualia* [14]. It requires sensory integration through multiple hierarchies and across motor systems, and it requires a brain. One of the problems with the “threshold of complexity” argument is that there is a consensus neither about how to measure the complexity of living systems [15] nor indeed about how to define a living system. In the latter case, ideas about the nature of the cell range from autopoetic systems [16] to tensegrity structures [17] and there is no consensus about what a cell actually *is* [18,19]. 

The “threshold of complexity” argument might be used in an attempt to disqualify bacteria on the basis of their size and their early origin in evolution; the counter-arguments are, respectively, that size and certain sorts of complexity are unrelated (or are even inversely related) and that their early origin has given bacterial complexity billions of years to evolve. A few authors have addressed the question of whether cells are conscious and have *q’.ualia* [5,9,20]. One of the popular candidates for the source of *qualia* is the microtubular network found in neurones [21] (however, see [22]). It may therefore be significant that tubulin has an equivalent in the FtsZ protein present in many (but not all) bacteria, that sonic communication between bacteria can be interpreted in terms of Frohlich-like phenomena [23], and that quantum entanglement has been reported in bacteria [24]. A relatively unexplored route to the hard problem is therefore to investigate whether bacteria are conscious, where consciousness in its most fundamental form is equated with subjective experience or *qualia*. 

The hard problem has been approached using *Integrated Information Theory*, which is based on taking the essential, phenomenological properties of *qualia* and then requiring them from the physical substrate [25]. According to Loorits, this hypothesis underpins the assertion that the hard problem has been solved because “*if* a phenomenon is analyzable in *fully* structural terms, then explaining the origin and nature of the structure of that phenomenon amounts to explaining the origin and nature of the phenomenon itself”, which is based on the idea that “the components of *qualia* are unconscious associations and the structures of *qualia* are the structures of networks of these unconscious associations” [26]. 

The inverse approach to the hard problem, which I undertake here, is to try to understand how living systems work by asking how they solve the most important problems that face them and then by asking how subjective experience might be involved. In the case of cells, it has been argued that one of these problems is how to generate, and how to proceed between, a very limited number of states or phenotypes when almost infinite choices seem to be available to them [27]. We have proposed that cells solve this problem of navigating phenotype space by selecting only a subset of the cell’s constituents to be *active* in determining its behaviour; this selection is achieved via *competitive coherence*, a scale-free concept that is based on the dynamics of patterns of connectivity in networks. We argue that this concept not only helps explain the behaviour of living systems at all levels but also captures the essential characteristics of living systems [28]. One of the advantages of competitive coherence as a concept is that its value as an organising principle can be demonstrated by simulating it as a neural net, which reveals both its ability to generate sequences of learned states (each corresponding to a subset of the system’s constituents) and the connectivity associated with these states [29,30]. Certain patterns of connectivity have been proposed as involving *qualia* [31] and, here, I take a leaf from *Integrated Information Theory* in proposing that it is the patterns of connectivity resulting from competitive coherence that reveal *qualia*. In the strong version of the competitive coherence hypothesis, *qualia* are not just associated with the generation of active states but are partly responsible for it; indeed, I further propose that the *qualia* at the highest levels of living systems are sufficiently strong so as to have a major effect on the functioning of these systems. In what follows, a program is used to simulate the operation of competitive coherence and thereby show not only how it permits the learning of a difficult task but also how it generates different patterns of connections that could characterise different *qualia*. I do not claim that this simulation actually generates *qualia* within the computer as, to quote Alfred Korzybski, “A map is not the territory it represents, but, if correct, it has a similar structure to the territory, which accounts for its usefulness”. Some neuroscientists and philosophers may be unfamiliar with just how sophisticated and important bacteria are and why it is interesting to study them in the context of *qualia*. This is not just because bacteria instantiate an experimentally accessible version of competitive coherence. It is also because bacteria: (1) exhibit complexity on many scales, (2) are closest in evolution to the first cells to appear on Earth, and (3) are inseparable from “higher” organisms with which they form holobionts. Moreover, bacteria and their accompanying viruses or bacteriophages are extremely abundant and make up a substantial proportion of the Earth’s biomass. Hence, there are good reasons to claim that bacteria are the dominant form of life on Earth and, in proposing that *qualia* have a function at the highest levels of organisation, I therefore propose that it is to bacteria we should turn. 

## 2. The Competitive Coherence Hypothesis

The hypothesis is made up of two parts. Firstly, competitive coherence characterises many living systems including bacteria. Secondly, competitive coherence generates *qualia*. 

### 2.1. Competitive Coherence

Biological systems on all scales are confronted with the challenge of how to be in a state that is coherent with both (1) present environmental conditions and (2) the system’s previous states [32]. These states correspond to the activity or functioning of an *Active* subset selected from all the constituents or elements of the system at a particular level of organisation. At the level of societies, organisations are constrained by the need to reconcile (1) coherence with their present environment and (2) coherence with their past states; for example, to grow and survive, research laboratories (1) must select an *Active* subset of PhD and post-doctoral students in response to new discoveries and to new funding initiatives but (2) must reconcile this selection with the research history of the laboratory and, in particular, with its historical skills, experience, and interests. There are a myriad of psychological, social and economic factors operating in this context but, from the point of view of a selection for the *Active* subset based on connectivity, these factors can be considered as links with different strengths. Note that connections between elements is an abstraction that subsumes all the interactions (physical, chemical, psychological, economic, etc.) between the elements of the system that affect whether an element becomes a member of the *Active* subset. 

At the level of brains, only a subset of neurones are active at any one time (depending on the quality of their firing rates and on excitatory and inhibitory inputs); if brain activity is to generate coherent behaviour, this *Active* subset must be related in a meaningful way to the previous subset (note that the size of this *Active* subset is important too since increasing it can result in a *grand mal* seizure). At the level of cells, a bacterium such as *Escherichia coli* has evolved to generate *Active* subsets in the form of cell states; as for other living systems, (1) a state must contain members that work coherently together and with the environment and (2) the states themselves must occur in a coherent sequence. This is because incoherence within and between cell states is counter-selected: a cell that simultaneously induces the genes for rapid growth in good conditions and for stress resistance in bad conditions is likely to be outcompeted by other cells; a cell that divides before replicating its DNA will die (see below). 

The concept of competitive coherence is that competition for two sorts of coherence underlies the generation of the *Active* subset of a system’s elements that determines the state of that system at a particular time [28,33,34]. I should stress that only systems that are associated with life exhibit competitive coherence. To grasp this scale-free concept, consider the problem of selecting an amateur football team each week from a larger group of potential players. Who plays this week is an important factor in determining who plays *next* week; this is a *Next* process at work. This is because, for example, it is easier to discuss shared transport arrangements with those already present than with those who are absent as they not playing that week. Hence, those who play this week are mostly likely to play again next week. Another important factor is the coherence of the team. Suppose there are two candidate goalkeepers available next week, the choice of one of them rather than the other may have consequences on the choice of other players since players must be chosen who can play together (and perhaps travel together); this is a *Now* process at work. In this analogy, the composition of the team each week is determined by a competition between the *Next* and *Now* processes so as to satisfy the demands of both the status quo and coherence.

Another example of competitive coherence, this time operating at a different level, occurs in the behaviour of individual animals. Consider a subordinate pack animal (i.e., not the alpha male) that has begun a series of behaviours that should result in “forbidden” mating; each of these behaviours corresponds to the loading into the *Active* subset via the *Next* and *Now* processes of the factors governing the behaviour. Suppose that this animal realises that the alpha male is racing towards it. This realisation corresponds to a *Now* process forcing the perception of the threat into the *Active* subset; the subsequent loading of the subset and resultant behaviour depend on a competition between a *Next* process to continue mating and a developing *Now* process to flee. 

It should be emphasized that the “competition” in competitive coherence is between two sorts of coherence, one reflecting the system’s history and the other its present situation. The connectivity-based competition for inclusion into the *Active* subset ensures that when the *Next* and *Now* processes are trying to load two different groups of elements into the *Active* subset (each resulting in a different behaviour), this subset does not contain some incoherent compromise. The importance of competition can be shown by a different principle—“collaborative” coherence—in which selection for inclusion in the *Active* subset is based on adding the *Next* and *Now* scores together for each element and then ranking them. Learning the simple task described below then becomes impossible (data not shown).

### 2.2. Competitive Coherence and Qualia

In the competitive coherence hypothesis, a coherent sequence of coherent states is fundamental to the functioning and indeed survival of living systems. As proposed above, each of these states is an *Active* subset of the elements of the system. Here, I propose that attaining each of these states at every level in the real world via competitive coherence corresponds to a *quale*, that is, a *quale* cannot be separated from the processes of attaining the state. Attaining these states is the result of a competition between the system’s elements. This competition is based on the connectivities of these elements in two networks, one, the *Next* connectivity, representing the history of the system, and the other, the *Now* connectivity, representing the present state of the system in its environment. Insofar as each *quale* corresponds to the generation of a particular state of a living system by competitive coherence, this means that *qualia* are intimately associated with particular patterns of connections and vice versa. 

To grasp this idea, it may be helpful to consider electric currents flowing in the same direction through two adjacent, parallel conducting wires; in this situation, the currents generate magnetic fields that tend to force the wires towards one another (though whether movement actually occurs depends on parameters such as the strength of the current). In an analogous way, when the two processes, the *Now* and *Next* processes operate together in competitive coherence (Figure 1), they generate *qualia* that tend to create the *Active* states. It will depend on the system as to whether the force exerted by a *quale* during competitive coherence is strong enough to help determine the *Active* state (see below). 

Each level of a living system, from the molecular to the planetary scale, has a corresponding complexity (that may or may not subsume the complexity of the levels below it) [29]. Competitive coherence operates at every level but operates with a complexity that depends on this level. In many living systems, there are many types of connection and hence many possibilities for forming the *Active* subset according to selection for these different types of connection. One possibility is to have an *Active* subset for each type of connection and then to have a second level of competitive coherence between the *Active* subsets themselves. In essence, each *Active* subset becomes an object in its own right and a second round of competitive coherence is performed on these objects using the connections between them; this corresponds to the transition to a higher level.

In the proposal here, the *qualia* that accompany the generation of the states via competitive coherence themselves become correspondingly complex and intense at higher levels such that, at the level of the planet, they have significant functions. These functions may result in actions on the material world or on something else. In which case, given the likely difficulty of denizens at one level in understanding those at a higher level, it is possible that the functions of *qualia* will remain forever obscure to us. 

## 3. Illustration of Competitive Coherence

To illustrate the hypothesis, a computer program was written based on competitive coherence [28] (see Appendix A). The essence of this program is a learning system comprising a set of elements of which small *Active* subsets are selected at different times that determine the state of the system in response to environmental inputs (Figure 1). Each element, **E**, which is identified by its address, has just two fields. These fields contain the addresses of the elements to which element **E** is connected. The *Now* field of element **E** contains the addresses of those elements that occur in the same state of the *Active* subset as element **E,** whilst the *Next* field of element **E** contains the addresses of those elements that occur in the state of the *Active* subset following a state in which element **E** has occurred. In Figure 1b, a system of competitive coherence is shown in which the *Active* state **S** comprises only three elements (**7**, **29** and **31**). Firstly, a *Next* process operates to count and rank the occurrences of the addresses in the *Next* fields of elements **7**, **29** and **31**; this results in the addresses at the top of the *Next* ranking being 14, 45 and 72. Since address 14 has the highest score, the corresponding element **14** is selected as the first member of the developing *Active* state, **S+1**. As element **14** (like all elements) has a *Now* field, a *Now* process can start to act. This *Now* process counts and ranks the occurrences of the addresses in the *Now* field of element **14**; this results in address 59 having the highest score and therefore being at the top of the *Now* ranking. Competitive coherence then entails a comparison between the *Next* score of address 45 (which is now at the top of the *Next* ranking) and the *Now* score of address 59 (which is now at the top of the *Now* ranking) and, since 45 has the higher score, element **45** is selected as the second element for the developing *Active*
**S+1** state. The *Now* process now counts and ranks the occurrences of the addresses in the *Now* fields of both element **14** and element **45** with the result that the score of address of element **59** increases (and remains at the top of the *Now* ranking). Competitive coherence then compares the *Next* address score of element **72** with the *Now* address score of element **59** and, because the address score of element **59** is higher, element **59** is selected as the final member of the *Active* state, which becomes the current state of the system. 

The program was adapted here firstly to run with a network of 100 elements and an *Active* subset of three of these elements; each element has a *Now* and a *Next* field; each field contains the addresses of a maximum of 30 different elements to which the element could be connected. The program was given the task of learning to respond to an input sequence of (1, 2, 3, 2, 3)_n_ with outputs (100, 99, 99, 100, 98)_n_, respectively. These inputs could be considered as representing different environments that require incompatible responses; for example, input 1 could represent a nutrient-rich environment suitable for growth with 100 as the appropriate response (production of the translational machinery), input 3 could represent a nutrient-poor environment unsuitable for growth with 98 or 99 as the appropriate responses (total or partial shut-down of the translational machinery), and input 2 could represent a transitional state with 99 or 100 as the responses (Figure 1a). 

To repeat, each input constitutes an element corresponding to an input (1, 2, or 3) being present in the *Active* subset of elements whilst each output constitutes an element corresponding to an output (98, 99 or 100) being present in a following state of the *Active* subset. A maximum of three states after the input is permitted before the program is forced to give an output. Learning depends on the changes in connectivity that result from the changes in the frequency of the addresses in the fields of the elements; for example, the contents of three of the network’s elements before and after learning are shown in Figure 2 (these three elements are chosen because they constitute one of the states of the *Active* subset shown in Figure 3). Before learning, the *Now* and *Next* fields of elements **5**, **15** and **16** contain a random selection of the addresses of the elements in the network whilst, after learning, these fields no longer have a random content: the *Now* field of element **5**, for example, mainly contains the addresses of elements **15** and **16** (Figure 2b). Put differently, the frequency with which a particular address occurs depends on whether the corresponding element has been part of the successful content of an *Active* subset; a reward process increases this frequency and a punishment process decreases it.

Competitive coherence is based on competition between the *Now* and *Next* processes for the selection of elements to be present in the *Active* subset. This selection depends on the number of times (equivalent to weights) the address of an element is present in the *Now* and *Next* fields of certain elements; the elements considered are those that have already been selected in the developing or in the preceding state of the *Active* subset, respectively. For example, to select progressively the elements for a new state of the *Active* subset (e.g., the **S+1** state in Figure 3a), firstly the *Next* fields of the elements present in the current state **S** (e.g., the elements **10**, **26** and **32**) are considered and the numbers of occurrences of the addresses in these fields are counted to give *Next* scores (e.g., the score of 31 for element **5** in Figure 3b); these scores are then ranked (Figure 3c) and the element with the highest *Next* score, here **5**, is then selected. Once there is an element in the *Active* subset, its *Now* field can be considered. In the case of element **5**, the highest *Now* score is 11 for the address of 15 (Figure 3d,e). There is now a competition between the elements with the highest *Next* and *Now* scores for selection for the **S+1** state of the *Active* subset, with selection going to the element with the higher scoring address; there are potentially two different candidates that could be selected, but here the same address, 15, has both the highest *Next* and *Now* scores. Element **15** is then selected. There are now two elements, **5** and **15**, in the *Active* subset and the addresses in the *Now* fields of both of these elements are scored and ranked. Again, the element corresponding to the higher of the scores of the addresses in the *Next* and *Now* fields, here **16**, is selected for the *Active* subset. 

There is a fundamental difference between the lists used by the *Next* and *Now* processes. The *Next* process relies on a ranked list of scores that, once calculated, does not change; if an element corresponding to an address at the top of the *Next* list is selected to be in the *Active* subset, the following address in the list replaces it at the top: the list itself does not change. The *Now* list, however, is recalculated every time an element is selected since the *Now* field of this new element must also be considered (by scoring its contents along with those of the *Now* fields of the other elements in the *Active* subset).

The nature of the competition between the *Next* and *Now* processes is easier to see in an example with a bigger *Active* subset, as in Figure 4, where this subset contains 13 elements out of the entire set of the 1000 elements in the system. The first seven elements (**495** to **251**) are selected from the *Next* scores (41 to 23, respectively) of their addresses because their scores are higher than the *Now* scores of the other elements. It is important to note that the *Next* scores in Figure 4a decrease from left to right as the corresponding elements are progressively selected for the *Active* subset whilst the *Now* scores increase from left to right as there are more and more addresses to be scored (which is because there are more elements in the *Active* subset and hence more *Now* fields). This change in the influence of the two processes results in an element from the *Now* process, **215**, being selected in preference to **40**, an element from the *Next* process because the address of element **215** has the higher score (Figure 4a).

The order of the elements contending for selection to the *Active* subset follows a clear pattern (Figure 4b). As elements are progressively selected from the *Next* ranking (which is obtained from the *Next* fields of the elements in the previous state of the *Active* subset), the curve of the scores of the elements in this ranking decreases whilst the curve of those from the *Now* fields increases. In the case of the *Next* field, this trend is simply due to the element with the highest score being selected first. In the case of the *Now* field, this trend is due to the pool from which the scores are obtained and increasing with each element selected. 

The overall connectivity of the network that accompanies the learning of a task by competitive coherence is reflected in the distribution of the frequencies with which the addresses of elements occur in the *Now* and *Next* fields of all the elements. Initially, the fields of all the 1000 elements were filled with the addresses of these elements chosen at random to give the set of all connections. Hence, a total of 30,000 addresses was chosen for the initial set (1000 elements x 30 addresses in each field). For each element, the number of times its address was present in the set was counted to give its rank (i.e., the number of connections in the set to that element). Then, the number of different addresses in the set with the same rank was calculated to give the distributions shown in Figure 5a,b for the *Now* and *Next* fields, respectively. These initial distributions, which are essentially binomial (although the three inputs and outputs were not connected to one another), changed as a result of learning and acquired a long tail (Figure 5c,d). In particular, the addresses of a small group of elements were present up to 80 times (i.e., had a rank of 80), and sometimes more, in the set of all connections.

In the competitive coherence hypothesis, the significance of these patterns of connectivity to *qualia* is that each pattern, like the one shown in Figure 4b, corresponds to the operation of a *quale*. In a *qualia*-driven living system, similar *qualia* (or sequences of *qualia*) may become connected to generate new perceptions and behaviours. The equivalent in the program would be for each pattern in the *Active* subset to be represented by one of a new class of elements that could then be linked to one another and selected via competitive coherence to join a separate, higher level *Active* subset. 

## 4. Bacteria and Competitive Coherence

How exactly does competitive coherence pan out at the level of a single bacterium? The bacterium *E. coli* can make use of a wide variety of different sources of carbon, nitrogen, phosphorus and sulphur to grow at a correspondingly wide range of rates, and growth in each of these different conditions is associated with a different set of cell states. *E. coli* has to contend with changes in oxygen tension, temperature, and osmotic and pH conditions, and may have to survive exposure to *uv*, desiccation or the presence of heavy metals, free radicals or antibiotics. To survive such vicissitudes, *E. coli* has many networks of genes controlled in part by transcriptional regulators of which there are over 300 [35]. *E. coli* can therefore produce a very large number of different cell states. In addition, even if the environment is constant and allows steady state growth, a vegetatively growing bacterium goes through the cell cycle, which comprises the events of the replication and the segregation of the chromosomes followed by division to give daughter cells. Each of these cell cycle events corresponds to a particular sequence of cell states. 

Bacterial cells have been selected with a capacity to grow and survive (as individuals and as populations). Put differently, bacteria have evolved to respond to both external and internal conditions. Such responses entail both the generation of a cell state, in which the cell’s contents work together efficiently and harmoniously, and the generation of a coherent sequence of cell states. Contradiction and incoherence are punished since, for example, a cell that simultaneously induces the expression of a set of genes encoding heat shock proteins and a set encoding cold shock proteins is likely to be outcompeted by rival cells that induce each set of genes only when needed. A similar fate would befall a cell that proceeded from one cell state to another very different one without good environmental reason; for example, cells that quit the spore state and germinate in hostile conditions are punished by death. There is therefore a strong selective pressure to evolve a mechanism for producing coherent cell states and for passing smoothly from one state to another state.

In the competitive coherence model of a simplified bacterial cell, the set of elements from which the *Active* subset can be selected include macromolecules (such as genes, mRNA, tRNA, rRNA, proteins) and small molecules and ions (lipids, polyphosphates, polyamines and polyhydroxybutyrates, ATP, CTP, cAMP, calcium, iron, zinc, etc.). A gene that it is being transcribed, an mRNA that is being translated, and an enzyme that is catalysing a reaction all belong to the *Active* subset. The bacterial equivalents of the *Now* and *Next* fields are the binding sites in the upstream regions of genes for transcriptional activators and repressors, terminators in mRNA, motifs in proteins that determine post-translational modifications (by kinases, methylases, proteases, etc.), monovalent and divalent ion binding, association with lipids, and association with other enzymes. The *Now* process corresponds, for example, to RNA and proteins that have affinities for one another assembling synergistically into a functional structure whilst the *Next* process corresponds to a transcription factor generated in one cell state leading to the transcription in the next cell state of the genes that depend on it. The number of cell states generated by competitive coherence would correspond, in the hypothesis, to the number of *qualia,* but this simplified picture of a bacterial cell is, however, just too simple (see Section 6). 

## 5. Bacteria and The Complexity Threshold

Subjective experience is sometimes considered to have a threshold of complexity below which an organism is “too simple” to have feelings. This might be thought to be the case of bacteria which were once believed to be simple, unstructured bags of enzymes. Now, we know that bacteria are complex and highly structured. Molecules and macromolecules often come together into spatially extended assemblies, *alias hyperstructures,* with specific functions [36] or *Self-Organize Whenever-And-Wherever-Needed* structures [37]. Hyperstructures are the descendants of the *composomes* proposed to have existed at the origins of life (see above). They include: cytoskeletal filaments similar to tubulin [38], actin [39], intermediate filaments [40], as well as EF-Tu [41,42], CTP synthase [43] and RNases [44] (which can form condensates [45]); condensates involved in signalling [46] and division [47], microcompartments for sequestering metabolic intermediates [48], “nucleoli” [49,50,51], chemoreceptor arrays [52], lipid rafts based on flotillin [53,54] or on cardiolipin [55], acidocalcisomes [56], polyphosphate granules [57], clusters of the E1 protein of the phosphotransferase system [58], the enzyme complexes of oxidative phosphorylation [59] and of other membrane proteins [60], proteins involved in DNA replication and segregation [61,62,63,64,65], “delay-regrowth” bodies [66], associations of genes, nascent RNA, and nascent proteins formed by the coupling of transcription, translation, and either insertion into membrane or into cytoplasmic complexes [67,68], and compartments for bacteriophages to evade host nucleases [69] (for additional references see [36,70]). It is worth noting here that bacterial cells are usually much smaller than the cells in many multicellular eukaryotes. This creates a greater constraint on bacteria to ensure as much processing as possible in as small a volume as possible (analogous to the density of components in a laptop compared with those in a mainframe computer). The result may be that bacteria are more complex per unit volume than neurones.

Bacterial complexity continues above the level of the single cell. Bacteria possess a wide variety of chemicals that they use to signal both to bacteria of the same and of different species and to their eukaryotic hosts [71]. These include molecules such as the broad-host range molecules that are used by bacteria to test population characteristics [72], or the molecules that interfere with other species [73] or the molecules that perturb their hosts [74,75,76,77]. Bacteria also communicate via physical mechanisms [78,79] that include sound [80] and electrical signals [81,82,83]. Some of this signalling can take place through nanotubes [84] and nanowires [85,86]. Quantum processes have been implicated in enzyme catalysis [87]. Intriguingly, in the context of signalling and organisation, some evidence for quantum entanglement at the bacterial level has been reported recently and attributed to a particular hyperstructure responsible for photosynthesis, the chlorosome [24]. 

It has been proposed that “there are two basic factors necessary to be considered a multicellular organism: cell–cell adhesion to form a new evolutionary unit, and intercellular communication leading to coordinated activity” [88]. The multicellular entities that bacteria create can be very complex. It has been proposed that there are three broad classes of bacterial multicellularity [88]. The first is that of filamentous bacteria, which are long chains of cells joined end-to-end that can be linear or branched and that can even share a cytoplasm. The second is that of the multicellular magnetotactic prokaryotes which assemble into ellipsoidal structures containing from 20 to 60 cells that appear to be connected by tight junctions. The third is that of bacteria that aggregate to create biofilms and swarms in which the cells are held together by an extracellular matrix (or by their flagella) and undergo differentiation; this class is sometimes considered to exhibit cooperation, altruism, exploitation prevention, kin discrimination, and allorecognition (the ability of an individual organism to distinguish its own tissues from those of another). 

There is still more to bacterial complexity than the above. Biofilms, for example, are not just multicellular but often comprise organisms from different species and even different kingdoms. They contain physiologically differentiated distinct subpopulations due to factors that include mutations and a multiplicity of different microenvironments due to the diffusion of intercellular signalling molecules, external stressors, nutrient/oxygen, and waste products; in turn, these microenvironments change due to feedback from the differentiated subpopulations. Hence, the biofilm grows and ages [89]. Biofilms, which can contain channels to transport nutrients and nanowires to conduct electrons, are believed to confer selective advantages of better resistance to stresses and better exploitation of nutrients [88]. 

## 6. Competitive Coherence and *Qualia* at Different Levels

If *qualia* are indeed generated by competitive coherence, this could occur at several levels of bacterial organisation and involve different players, different types of connections and different processes. In the case of genetic organisation at the hyperstructure level, competitive coherence affects a particular hyperstructure as follows: a *Next* process allows those genes that are already expressed as part of a hyperstructure to help determine which genes are expressed next in that hyperstructure [30]. A *Now* process allows those genes that are starting to be expressed together in a hyperstructure to recruit related genes to the hyperstructure. 

In the case of organisation at the level of the cell, the state of a cell at any one time corresponds to the set of hyperstructures present within it. The hyperstructures in these *Active* subsets would usually include those for ribosome synthesis, chemotaxis, ATP generation, sugar metabolism and transport, and cell cycle events (see above). At this higher level (as with the generation of a hyperstructure at the lower level), the composition and existence of the cell result from competitive coherence between hyperstructures in which a *Next* process ensures continuity and a *Now* process ensures coherence. A new cell state is the result of (1) a *Next* process, whereby the current set of hyperstructures in the cell determines the next set, and (2) a *Now* process, whereby the developing set of hyperstructures progressively recruits, maintains or dismisses hyperstructures. Competition between these two processes ensures a sequence of sets of hyperstructures (cell states) that optimise growth and/or survival.

Competitive coherence also determines the functioning of systems at levels above the cell. In the case of bacteria, these include the biofilm (where the *Active* subset corresponds to the cells that are growing or moving) and the microbiome within a holobiont (where the *Active* subset corresponds to the cells that are growing and signalling to one another and to the host [90]). Competitive coherence could also be invoked in the case of the species within an ecosystem (where, out of all the species present, the *Active* subset corresponds to the species that are determining the fate of other species and of the environment) and even ecosystems within the living planet (where an *Active* subset of ecosystems exists out of all those that might exist). 

The fundamental question here is whether the importance of *qualia* in the functioning of a living system increases with that level of that system such that *qualia* only play a significant role at the highest levels. As the states generated by competitive coherence become increasingly complex with the increase in the level of the system, the accompanying *qualia* become correspondingly complex and influential. At some level, these *qualia* become functionally analogous to the role played by gravity in the organisation of matter, which becomes increasingly important as mass increases (for the sake of symmetry, the possibility should also be considered that *qualia* are only of significance at the lowest levels of organisation, that is, the sub-atomic). This level may be that of the cell, the collection of cells (like the brain), the ecosystem or the planet (or *qualia* may act on all of them in different ways). In the case in which the actions of *qualia* are at a level above that of individual human beings, it is likely that they will remain forever inexplicable, just as the function in perception of the passage of an ion flux in a channel in a neurone is inexplicable to that channel. 

## 7. *Qualia* and The Bacterial Origins of Life

Where do *qualia* come from? One possibility is that *qualia* are inseparable from the material world [91], in which case they would have been present in the first cells. A second possibility is that *qualia* are only present in living systems; this could be either because living systems are doing something that generates *qualia* (such as selecting, storing and recalling information) or because these systems are constructed in a certain way out of certain materials (such as having cytoskeletal structures made of tubulin [92]). For either possibility, an important question is whether the stage of evolution—appearance of the first cells, appearance of the first eukaryote, etc.—is relevant to the use a living system could make out of *qualia*. The answer to this question depends on the nature of life at its origins and, because bacteria were close to these origins, it is reasonable to suppose that it involves bacteria. 

The hypothesis that microbes actually initiated life on Earth via *panspermia* [93,94] is controversial. It is, however, generally agreed that, over three billion years ago, cyanobacteria did play a fundamental role in creating our atmosphere via oxygenic photosynthesis [95] and in fixing nitrogen [96]. In the “prebiotic ecology” approach to the origins of life, a flux of abiotic creation and destruction led to the preservation and accumulation of the molecules that interacted to form spatially extended assemblies or composomes [97,98,99]. These composomes (the precursors of hyperstructures) grew and divided but, in each composomal lineage, retained an identity [98]. Competitive coherence operated during the growth and evolution of composomes insofar as new molecules were recruited to a composome on the basis of the molecules already present (which is a *Next* process) but some of these new molecules had their own, different preferences for other molecules (which is a *Now* process). 

## 8. Bacteria, Dominant Species and Holobionts

There is a flaw in the neurocentric reasoning that since humans are the dominant species on Earth due to the functioning of their brains they therefore possess consciousness in its highest form. This is because, by many criteria, bacteria were, are, and will be the dominant species on Earth. They were here first, they occupy almost every possible niche from over a kilometre deep in the crust [100] to the upper troposphere [101], and, given the likely capacity of archaea and bacteria to survive in suspension in salt crystals for geological times [102,103], they will be here long after we are extinct [104]. They make up much of the Earth’s biomass though it is unclear exactly how much [105]. Indeed, it has been estimated that the Earth harbours over 10^30^ bacteria [106] and possibly 10^31^ bacteriophages, *alias* bacterial viruses [107]. These bacteriophages allow the transfer of genetic material between bacteria via transduction, a method that adds to the arsenal of other mechanisms of genetic transfer such as conjugation and transformation. It has therefore been proposed that bacteria and bacteriophages form a single, global, superorganism [108]. A superorganism can behave as an entity in its own right. The simulation of a swarm of bees, *Apis mellifera*, as a superorganism shows that its behaviour resembles that of individual organisms insofar as the swarm obeys psychophysical laws, which include the relationship between the length of time taken to choose a nest site and the number and quality of the sites available [109]. By one of the criteria of our sophisticated species, bacteria might actually be considered as superior to humans insofar as we are able to eliminate few bacterial species whilst they kill millions of us every year and, as they are so adept at solving problems, will continue to develop antibiotic resistances to kill even more in the future [110].

Even if subjective experience were confined to “superior” animals such as humans, we would need to take into account what exactly these animals are. Rather than existing as an independent organism, a multicellular host and its microbiota constitute a *holobiont* [111,112], which is the sum of all the different genomes present in a eukaryotic organism. In other words, a holobiont comprises the host genome and the genomes of the symbiotic microbiota; hence, a holobiome can be considered as the genomic reflection of the complex network of symbiotic interactions that link an individual member of a given taxon with its associated microbiome.

We too are holobionts. In (and on) a human, bacterial cells outnumber human cells even if the total mass of these bacteria is much less than that of the human cells [113]; moreover, from the point of DNA, these bacteria have around nine million different genes compared with only 20 thousand different human genes [114]. Our resident bacteria help determine our serum metabolome [115], manipulate our immune systems [76,116,117], alter our susceptibility to many diseases, including cancer [118,119,120], heart disease [121], affect ageing in different species [122,123], including age-related macular degeneration [124], and modify our responses to drugs [125]. They also influence our behaviour [90,126,127,128,129,130] (note though that caveats have been raised in the case of rodent-based studies [131]). It is in such influences on behaviour that a relationship between bacteria, hosts and *qualia* can be discerned. For example, those bacteria in the gut that benefit most from their hosts consuming a particular food would have a selective advantage if they could make such consumption a pleasure for the host [90,132]; reciprocally, one might argue that a selective pressure exists for a bacterial species to make its host averse to the food that feeds rival species.

## 9. Discussion

The connectivity of many types of systems can be described using concepts such as *small world networks* and *self-organised criticality* [31,133,134,135]. The concept of competitive coherence can also be used to describe the connectivity of a system but is special because, in selecting the *Active* subset of the elements of the system that determines its behaviour at a particular time (i.e., its state), competitive coherence reconciles the coherence of behavioural states over time (i.e., its history) with the coherence of the present state (i.e., the coherence of the state’s elements with one another and with the environment). The fact that only living systems have been selected—or have learnt—to perform competitive coherence means that the connectivity generated by competitive coherence may have characteristics unique to life. Some of these characteristics can be captured by training a neural net based on competitive coherence to learn a task. From this program, the connectivities of the *Next* and *Now* elements can be obtained as they are recruited to the *Active* subset. These results show that competitive coherence generates states that, in terms of connectivity, have structured, differentiated and unified properties. Such states are required of *qualia* by *Integrative Information Theory* [25]. In this kind of approach, “the ability to repeatedly recognize their own representations in cycles of local feedback could lead to the generation of *qualia*” [136]. Recognising its own representation would correspond to the living equivalent of the competitive coherence program recognising a particular pattern of connections (e.g., the *Now* and *Next* curves shown in Figure 4b) and then linking it to similar patterns. Of course, this is a toy model and, in reality, the sizes of the *Now* and *Next* fields of each element would vary with the importance of the element in the task that has been learnt whilst the size of the *Active* subset would also vary with, for example, the growth of the system.

The idea that *qualia* result from competitive coherence to play a role in the functioning of a living system echoes the idea that consciousness plays a role in the functioning of an internal integrating system concerned with representational operations [137]. Fundamental experiences have been considered as the basis of causation (for references, see [2]) and *qualia* are considered as a causative factor in my hypothesis too, albeit only at a very high level of living systems (see below). This sounds as if I am saying that *qualia* are associated with a system when it reaches a certain level of complexity. This idea has been criticised on several grounds [138]. It should be noted that as my proposal is that *qualia* depend on the operation of competitive coherence, which is quantifiable [29], the problem of defining complexity is avoided [138]. Moreover, the problem of how to define the system exhibiting consciousness is also circumvented [138] since a living system is defined by a frontier of discontinuity in the connectivity of elements [19] (for references, see [139]) and it is on such connected elements that competitive coherence operates.

It is particularly interesting to consider bacteria when exploring the possible relationship between competitive coherence, life and *qualia*. This is because bacteria were fundamental to the origins of life and, according to several criteria, are the dominant lifeform on Earth. Where then should one look in the bacterial world for relationships between competitive coherence and *qualia*?

At the intracellular level, bacteria have a rich, hierarchical and “hypercomplex” organisation that comprises assemblies of molecules and macromolecules termed hyperstructures [29]. Insofar as the formation of a hyperstructure from its elements is determined by competitive coherence, a *quale* might accompany or indeed drive such formation. This could be the case, for example, in the assembly of the hyperstructures in which the ribosomes (which make all the cell’s proteins) themselves are made; the richer the growth medium, the larger these hyperstructures. Assuming that pain and pleasure are the most fundamental *qualia*, the *quale* corresponding to the assembly of a large ribosomal hyperstructure might be one of contentment or satiety.

At a higher level of the individual bacterium itself, there are three areas in particular where *qualia* might be sought since they all involve major changes in connectivity. Firstly, chemotaxis in bacteria such as *E. coli* entails cells moving up a concentration gradient of an attractor or down a gradient of a repellent, swimming continuously whilst the concentration increases or decreases, respectively, and tumbling (to find a new direction) when the concentration ceases to improve [140]; this behaviour is mediated by a series of states of hyperstructures [52]. It is conceivable that continuous swimming is accompanied by pleasure and that tumbling is accompanied by pain. Secondly, the lytic cycle of a bacteriophage entails the bacteriophage multiplying and destroying the bacterium. It might be imagined that the bacterium undergoing such lysis experiences pain. Thirdly, the cell cycle in bacteria such as *E. coli* entails major changes in its hyperstructures to accompany and drive the replication and segregation of its DNA and the subsequent cell division that gives two daughter cells. It has been proposed that initiation of chromosome replication in this bacterium as it grows is related to the changes in hyperstructure dynamics that it would experience if such replication were postponed; in this case, the *quale* corresponding to attaining the maximum size of the ribosomal hyperstructure might be the feeling of satiety, whilst the *quale* corresponding to the assembly of the replisome might be one of excitement. Moreover, cell division is a risky process and involves the assembly of the divisome and its action on the cell wall, which puts the cell at risk of lysis, and in this case the *quale* might be anxiety. These *qualia*, along with others, might form part of the overall cell cycle experience at the level of the bacterium itself.

At the level of large bacterial populations, it is instructive to compare them with the ensembles of neurones that make up brains. Bacterial cells contain similar macromolecules to those found in neurones and these macromolecules exist in similarly structured intracellular environments. Whilst neuronal connections are limited to synapses on dendrites, bacterial connections are very diverse and include nanotubes [84] and nanowires [85,86] as well as sonic [80] and electromagnetic signalling [81,82,83]. An association between *qualia* and the activities of these bacterial populations is no more absurd than a similar association with the activities of collections of neurones.

In the context of competitive coherence, what—if anything—might be the functions of *qualia*? At the level of interactions between bacteria, it might be imagined that the pain experienced by a bacterium as it undergoes lysis by bacteriophages is sensed by other bacteria, but I know of no evidence that resistance to bacteriophages can be transmitted other than genetically [141]. In the case of *qualia* and cell division, again, I know of no evidence that division in one cell affects division in others. In the case of chemotaxis and other organised movements, a function for *qualia* is conceivable given that *E. coli* cells can move collectively as a swarm [140]. At still higher levels of organisation, I favour the hypothesis that the nature of *qualia* changes to become increasingly complex and intense. This results in high-level *qualia* having significant functions. In these functions, bacteria are likely to play a major role since they constitute a substantial part of life at the levels of holobionts, ecosystems and planet Earth. The task of understanding such high-level functions may fall into the domain of theologians but risks being unknowable.

## 10. Conclusions

Of all the ways of generating connections, competitive coherence is special because its operation is characteristic of living systems at all hierarchical levels. The fundamental proposal of this paper is that the particular dynamics and structures of connections involved in competitive coherence are inseparable from *qualia*. *Qualia* are generated by—and can help generate (in higher level systems)—the competitive coherence that determines the state of a living system. Indeed, the different patterns of connectivity generated by competitive coherence are the connectivity correlations of different *qualia*.

## Figures and Tables

**Figure 1 biology-10-01034-f001:**
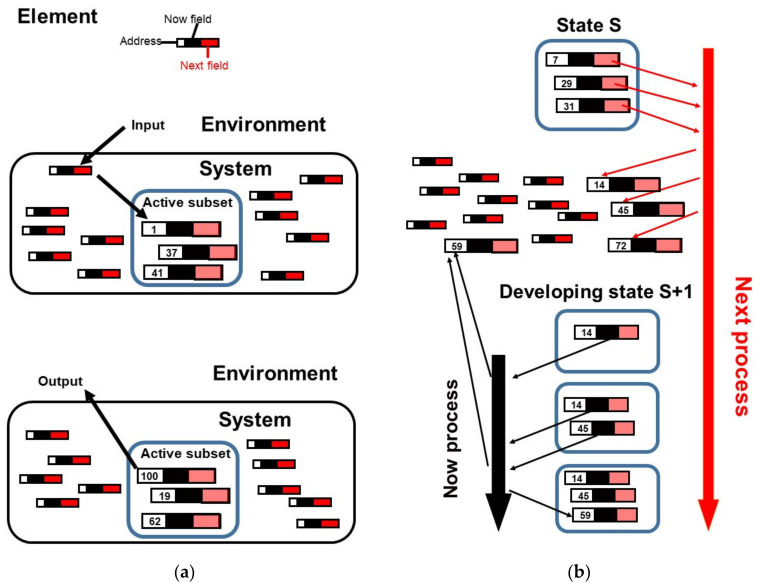
Simulation overview of competitive coherence. (**a**) A living system comprises a set of elements. Each element has an address along with a *Now* field and *Next* field that contain the addresses of the other elements to which it is connected. A subset of the elements is selected to belong to the *Active* state of the system; this subset determines the behaviour of the system. The environment acts on the system by causing the appropriate input elements (here element **1**) to be loaded to the *Active* subset. The system acts on the environment by causing the appropriate output elements (here element **100**) to be loaded to the *Active* subset. (**b**) The *Active* state **S**, the present state of the system, contains three elements (**7**, **29** and **31**). To generate the next
*Active* state, **S+1**, the connections in the *Next* fields of these elements are scored by counting the number of times each address figures in these fields; the elements with the highest *Next* address scores are, in order, **14**, **45** and **72**. Element **14** is then selected to be the first member of the developing *Active* state, **S+1**. Element **14** has a *Now* field containing the addresses of the elements that are frequently found with it in the same *Active* state; element **59** has the highest *Now* address score. Selecting the second element for the developing *Active* state depends on comparing the *Next* address score of element **45** with the *Now* address score of element **59**. Here, element **45** has the higher address score and is therefore selected. The addresses in the *Now* fields of both elements **14** and **45** are then scored; element **59** again has the highest score. Selecting the third (and here the final) element for the developing *Active* state **S+1** entails comparing the *Next* address score of element **72** with the *Now* address score of element **59**. Here, element **59** has the higher address score and is therefore selected. *Active* state **S+1** becomes the present state of the system. Rectangles represent the elements; the first number is the address of the element, the black part is the *Now* field and the red part is the *Next* field; the rectangles are bigger when the element is included or is likely to be included in the *Active* state. The large rounded rectangles represent *Active* states.

**Figure 2 biology-10-01034-f002:**
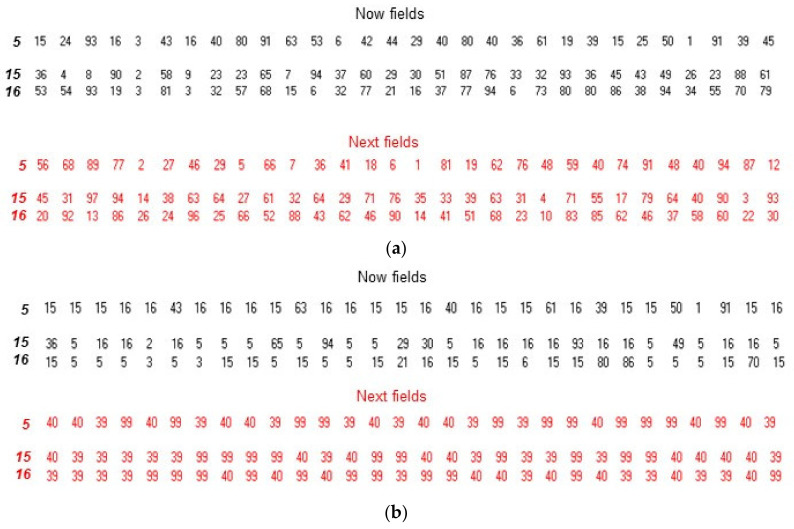
The connectivity of the elements is altered by learning a task. In a toy system in which Table 100. elements, elements **5**, **15** and **16** constituted one of the states of the *Active* subset. Each of these elements is connected to other elements by the frequency of the addresses of these other elements in the *Now* and *Next* fields of **5**, **15** and **16**. (**a**) Before learning, elements **5**, **15** and **16** (in bold italics) have fields that contain a random selection of the addresses of all the elements. (**b**) After learning, the contents of these fields are no longer random: the *Now* fields mainly contain the addresses of the other two elements that are found in the same state of the *Active* subset whilst the *Next* fields mainly contain the addresses of the three elements (**99**, **39** and **40**) that are found in the following state (i.e., the *next* state) of the *Active* subset. *Now* fields are in black and *Next* fields are in red.

**Figure 3 biology-10-01034-f003:**
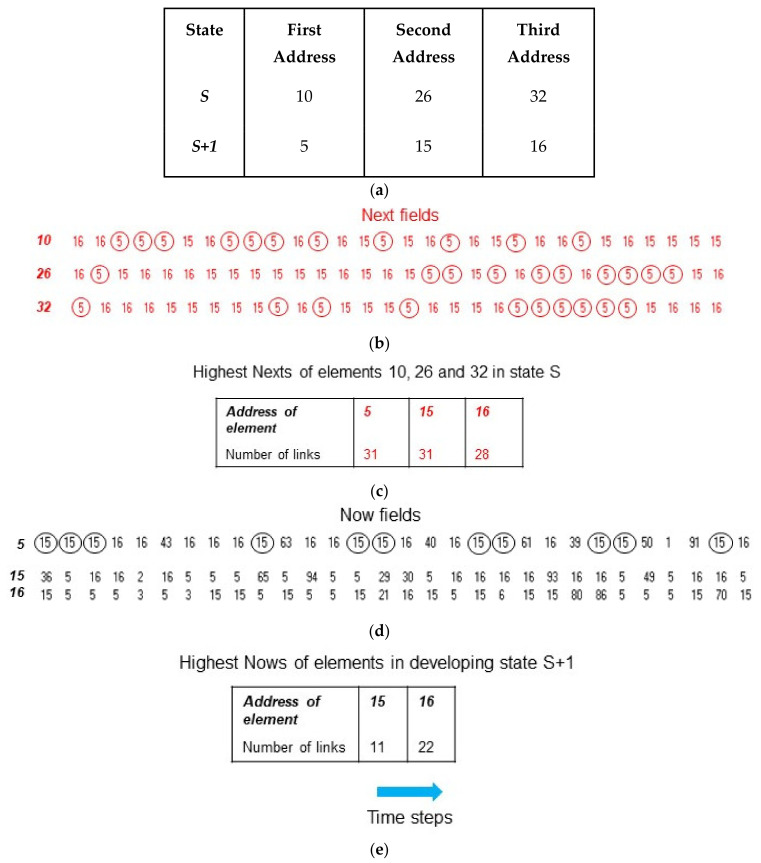
Generating states of the *Active* subset. After learning based on competitive coherence, the toy system here has 100 elements of which three are selected at any one time to form the *Active* subset. (**a**) Two successive states of the *Active* subset showing the order with which the elements are selected. (**b**) The contents of the *Next* fields of the elements in state **S** of the *Active* subset with the occurrences of one of these addresses (5) circled. (**c**) The addresses shown in (**b**) are counted and ranked, with the score of the address of element **5** being equal highest with that of element **15**. (**d**) Element **5** is selected for the *Active* subset with the occurrences of one of the addresses in its *Now* field (15) circled. (**e**) The *Now* addresses are counted and ranked. The score of the address of element **15** is highest in both the *Now* and *Next* rankings and element **15** is therefore selected as the second element in the developing **S+1** state of the *Active* subset; the *Now* fields of both elements **5** and **15** are then scored and ranked, with the address of element **16** having the highest score; since the score of this address is highest in both the *Now* and *Next* rankings, element **16** is selected as the third element in the **S+1** state of the *Active* subset.

**Figure 4 biology-10-01034-f004:**
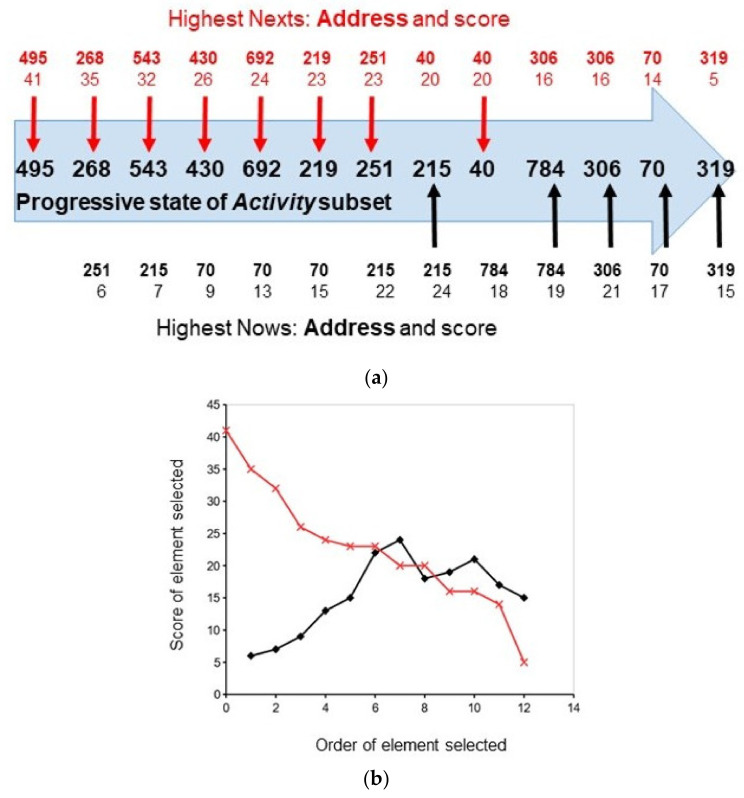
The connectivities of the *Now* and *Next* processes. The simulated system has an *Active* subset of 13 elements that are selected from a total of 1000 elements. (**a**) The temporal order of the elements selected for the *Active* subset is shown by the blue arrow. Firstly, element **495** is selected because its address has the highest score (41); secondly, element **268** is selected because its address in the *Highest Next* ranking has a score (35) higher than that of the address of element **251** in the *Highest Now* ranking (6); the importance of the *Now* process relative to the *Next* process increases as more elements are selected and the 8th element selected, **215**, is preferred to element **40** because the former’s *Now* address score is greater than the latter’s *Next* address score; after the 9th position in the selection, the *Now* process dominates. (**b**) A graph showing the relationship in the progressive selection of elements for the *Active* subset between the scores of the addresses at the top of the *Now* (black) and *Next* (red) rankings.

**Figure 5 biology-10-01034-f005:**
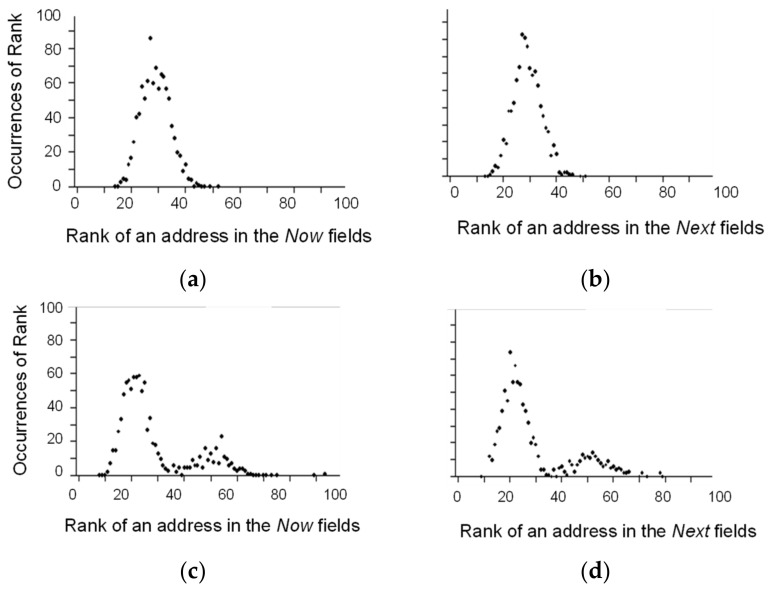
The changes in the connectivity of the entire network following learning. The addresses present in the 30000 *Now* and *Next* connections in the network (1000 elements x 30 addresses in each field) were ranked according to how many times they appeared. Then, the number of different addresses in each rank were plotted against the rank for: (**a**) The *Now* fields before learning (**b**). The *Next* fields before learning (**c**). The *Now* fields after learning (**d**). The *Next* fields after learning.

## Data Availability

Not applicable.

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
