# Peer review of "Competitive Coherence Generates Qualia in Bacteria and Other Living Systems"

_biology, 2021, doi:10.3390/biology10101034_

Round 1
Reviewer 1 Report
see attached file.

Reviewer 2 Report
Review of Norris for Biology
Norris's core thesis, that bacteria exhibit the behaviors and have the biological complexity for qualia, is one I am in agreement with as I have argued for a similar resolution of the "Hard Problem" of consciousness.
I am somewhat less comfortable with the general framework, borrowed as noted, from IIT a model developed by Tononi and colleagues, that focuses on general mechanisms that cite complexity (in some form) as the epistemic cause(s). Norris presents his "competitive coherence" (CC) model as an in principle, complete characterization of the underlying mechanisms that give rise to qualia. And to make this proposal "go though" he notes that bacteria display all of the necessary qualities. So, the obvious question is: "Do prokaryotes have qualia because their biomolecular functions display the requisite competitive coherence or do they manifest the kinds of competitive coherence because they have the requisite biomolecular mechanisms that instantiate sentient organisms?"
This isn't a trivial distinction and, ultimately, it rests on the question of whether an artificial, non-biological entity that had the necessary (and presumably sufficient) level of competitive coherence would be sentient. Norris is agnostic here but seems to side with the conclusion that the system must be biological and not artificial though, as noted, Tononi's IIT theory tilts the other way.
Norris's effort at showing how such a system could produce qualia, the computer program described in Section 3 (beginning at line 194) is, in my mind, unpersuasive. Its code is written so that the elements of CC are present but there's no consciousness, no qualia − just a (rather confusing) description of what might be going on in a bacterium that makes it sentient. If the paper is accepted, I recommend either shortening this section or removing it completely.
The section arguing for the dominant role of bacteria is similarly misplaced. While superficially correct, it has no relevance to the core arguments about the mechanisms for sentience in prokaryotes.
My recommendation is to publish with the two suggested edits.
